# Epidemiology of Chronic Kidney Disease in Children: A Report from Lithuania

**DOI:** 10.3390/medicina57020112

**Published:** 2021-01-26

**Authors:** Jūratė Masalskienė, Šarūnas Rudaitis, Renata Vitkevič, Rimantė Čerkauskienė, Diana Dobilienė, Augustina Jankauskienė

**Affiliations:** 1Department of Children Diseases, Medical Academy, Lithuanian University of Health Sciences, LT-44307 Kaunas, Lithuania; sarunas.rudaitis@kaunoklinikos.lt (Š.R.); diana.dobiliene@kaunoklinikos.lt (D.D.); 2Vilnius University Hospital Santaros Klinikos, LT-08406 Vilnius, Lithuania; renata.vitkevic@santa.lt (R.V.); rimante.cerkauskiene@santa.lt (R.Č.); augustina.jankauskiene@santa.lt (A.J.); 3Pediatric Center, Vilnius University, LT-08406 Vilnius, Lithuania

**Keywords:** children, chronic kidney disease, epidemiology, renal replacement therapy, treatment

## Abstract

*Background and Objectives:* The data on the prevalence of chronic kidney disease (CKD) in the pediatric population are limited. The prevalence of CKD ranges from 56 to 74.7 cases per million of the age-related population (pmarp). The most common cause of CKD among children is congenital anomalies of the kidney and urinary tract (CAKUT). With progressing CKD, various complications occur, and end-stage renal disease (ESRD) can develop. The aim of the study was to determine the causes, stage, prevalence, and clinical signs of CKD and demand for RRT (renal replacement therapy) among Lithuanian children in 2017 and to compare the epidemiological data of CKD with the data of 1997 and 2006. *Materials and Methods:* The data of 172 Lithuanian children who had a diagnosis of CKD (stage 2–5) in 1997 (*n* = 41), in 2006 (*n* = 65), and in 2017 (*n* = 66) were retrospectively analyzed. Physical development and clinical signs of children who had CKD (stage 2–5) in 2017 were assessed. *Results:* The prevalence of CKD stages 2–5 was 48.0 pmarp in 1997; 88.7 in 2006; and 132.1 in 2017 (*p* < 0.01). Congenital and hereditary diseases of the kidney in 1997 accounted for 66% of all CKD causes; in 2006, for 70%; and in 2017, for 79%. In 2017, children with CKD stages 4 or 5 (except transplanted children) had hypertension (87.5%) and anemia (50%) (*p* < 0.01). Children under ≤2 years with CKD were at a 3-fold greater risk of having elevated blood pressure (OR = 3.375, 95% CI: 1.186–9.904). *Conclusions:* There was no change in the number of children with CKD in Lithuania; however, the prevalence of CKD increased due to reduced pediatric population. CAKUT remains the main cause of CKD at all time periods. Among children with CKD stages 4 or 5, there were more children with hypertension and anemia. In children who were diagnosed with CKD at an early age hypertension developed at a younger age.

## 1. Introduction

Chronic kidney disease (CKD) is a major public health issue: 11–13% of the overall world’s population suffers from this disease [1]. The World Health Organization (WHO) has recently added kidney and urologic disease to the mortality information tracked worldwide and this should be a valuable source of such data over time despite the WHO not posting the information by age groups [2]. In the literature, the data on the prevalence of CKD among children are scarce. According to the data of 2007, the prevalence of pediatric CKD ranged from 15 to 74.7 cases per million of the age-related population (pmarp) [3]. Such a wide range in the prevalence of CKD was caused by the lack of agreement on the unified diagnostic and classification system for CKD for a long time. The Kidney Disease: Improving Global Outcomes (KDIGO) guidelines released in 2002 and updated in 2012 proposed the definition and classification of CKD and chronic renal failure (CRF) as well as provided recommendations how to delay disease progression and prevent complications [4].

The prevalence of pediatric CKD in the European Union (EU) countries ranges from about 55–60 to 70–75 pmarp in Spain and Italy, depending on the clinical definition of CKD that was used in each study [5]. In Lithuania in 1997, the prevalence of CRF was 48.0 pmarp [6], and in 2006 this number almost doubled, i.e., increased to 88.7 pmarp [7]. Congenital anomalies of the kidney and urinary tract (CAKUT) constitute the most common cause of pediatric CKD (48–59%) [2]. With advancing CKD, various complications occur, and their frequency depends on the stage of CKD. Anemia is the most common complication with the prevalence being 73% at stage 3 CKD and >93% at stage 5 CKD [8]. Growth impairment, occurring in 35% of children with CKD stages 2–4, is the most difficult to manage and the most psychological challenges-causing issue [9]. According to Harambat et al., 43% of patients with childhood onset end-stage renal disease (ESRD) do not achieve an adult height within the normal range [10]. Moreover, the actual prescription of recombinant growth hormone among growth-retarded ESRD children is low [11]. In part of the children, CKD gradually progresses to ESRD when renal replacement therapy (RRT) is applied. According to the data by the European Society for Paediatric *Nephrology* and European Renal Association - European Dialysis and Transplant Association (ESPN/ERA-EDTA) registry, the prevalence of pediatric RRT in 2012 in Europe was 27.9 pmarp; in Lithuania, it was 22.2 pmarp [12].

The epidemiology of chronic renal failure in children was last examined in Lithuania in 2006. Since then, the classification of the disease has changed, and patients’ care and treatment have improved. Therefore, the aim of this study was to identify the causes, stages, prevalence, and clinical signs of CKD and the demand for RRT among Lithuanian children in 2017 and to compare these epidemiological data on CKD with those of 1997 and 2006.

## 2. Materials and Methods

The data of 172 Lithuanian children who had a diagnosis of CKD on 31 December 1997 (*n* = 41), 31 March 2006 (*n* = 65), and 1 January 2017 (*n* = 66) were retrospectively analyzed. The study was carried out at the Children’s Hospital, Affiliate of Vilnius University Hospital Santaros Klinikos, and the Department of Children Diseases, Hospital of the Lithuanian University of Health Sciences, as all such children are followed up and treated in these healthcare institutions. The data for analysis were collected by using electronic patient databases of these institutions.

The study enrolled children with CKD stages 2–5. In 1997, the age of children ranged from 1 month to 16 years, and in 2006 and 2017, from 1 month to 18 years. The glomerular filtration rate (GFR) of all children included in the study was <90 mL/min/1.73 m^2^ for three months before the study. The GFR was calculated by using the Schwartz formula [13]. In 1997, the degrees of CRF were defined based on the following criteria: grade 1, GFR 41–80 mL/min/1.73 m^2^; grade 2, GFR 21–40 mL/min/1.73 m^2^; and grade 3, GFR ≤ 20 mL/min/1.73 m^2^. CKD stages were defined based on the published 2012 KDIGO guidelines [4]. Following these guidelines, CRF corresponds to CKD stages 2–5. CKD stage 5 was defined as ESRD when a child needed RRT. Children’s age and gender, cause and prevalence of CKD, and frequency of RRT were analyzed, and the data obtained were compared by the year. Physical development and clinical signs of children who had CKD stages 2–5 on 1 January 2017 were assessed. Standard deviation scores (SDS) for height were calculated according to recent national growth charts whenever available or according to the recently developed Northern and Southern European growth charts [14]. Height was evaluated using sex- and age-adjusted percentiles. Height was considered normal when it fell within the range between the 3rd to the 97th percentile; height below the 3rd percentile was deemed short stature. Blood pressure was measured with a mercury sphygmomanometer using a cuff appropriate for the child’s age. Hypertension was diagnosed following the published 2016 European Society of Hypertension guidelines for the management of high blood pressure in children and adolescents [15]. Anemia, secondary hyperparathyroidism, and acidosis were defined based on the KDIGO guidelines [4].

The study was approved by Vilnius (No. 158200-18/5-1039-535) and Kaunas (No. BE-2-62, 3 September 2018) Regional Biomedical Research Ethics Committees.

Statistical analysis was performed using the SPSS 22.0 software (SPSS, Chicago, IL, USA). Distribution of continuous variables was assessed with the Kolmogorov-Smirnov test. Continuous variables were compared using the parametric Student’s *t*-test or the nonparametric Mann-Whitney tests, and for the comparison of more than two groups, analysis of variance (ANOVA) or nonparametric Kruskal-Wallis test was used. Categorical data were compared with the chi-squared (*χ*^2^) test. Depending on the sample size, the exact test (for small samples) and the asymptotic *χ*^2^ criterion were used. Receiver operating characteristic (ROC) curves by calculating the area under the curve (AUC) were used to evaluate the diagnostic discrimination and cut-off value of age at the diagnosis of CKD to predict hypertension.

*p*-values < 0.05 were considered statistically significant.

## 3. Results

### 3.1. Epidemiology of Chronic Kidney Disease

In 1997 in Lithuania, there were 41 children with CKD stages 2–5 [6]; in 2006, 65 [7]; and in 2017, 66. Children with CKD stages 2–5 in 1997 accounted for 48.0 pmarp; in 2006, for 88.7 pmarp; and in 2017, for 132.1 pmarp (*p* < 0.01). There was no significant difference in the mean age of the study population by year (Table 1). At all timepoints, the male predominance was observed (Table 1); the ratio of girls to boys in 1997 was 1:1.05; in 2006, 1:1.17; and in 2017, 1:1.32.

In 1997, the patients were stratified by CRF grade: grade 1, 15 (36.6%); grade 2, 10 (24.4%); and grade 3, 16 (39%) children, where CRF grades 1 to 2 correspond to CKD stages 2–4.

Children with CKD stage 2 accounted for the largest proportion in 2006 and 2017. Based on the CKD criteria, 61% of the examined children had CKD stages 2–4 in 1997; 75% in 2006; and 78% in 2017.

In 1997, there were 16 children with ESRD and this accounted for 8.7 pmarp. Ten (62.5%) children lived with a transplanted kidney. In 2006, 16 children were diagnosed with ESRD (21.8 pmarp), in 2017—14 children (60.0 pmarp). Ten children (71.4%) in 2017 had a transplanted kidney (*p* < 0.05).

CAKUT was the main cause of CKD among Lithuanian children, and there was an insignificant increase in its prevalence with years. The proportion of children with cystic kidney disease was decreasing, while that with hereditary nephropathies was increasing (*p* < 0.05) (Table 2). Congenital and hereditary diseases of the kidney occurred more frequently with years, and they made up 66% of all CKD causes in 1997; 70% in 2006; and 79% in 2017.

### 3.2. Epidemiology of Chronic Kidney Disease in 2017

On 1 January 2017, 66 children had CKD stages 2–5. Their distribution by CKD stage, gender, mean age, and age when a CKD-related cause was identified did not differ statistically significantly (*p* > 0.05) (Table 3). CKD stage 5 was diagnosed in 21.2% of the study population. Boys accounted for 57.7%. The mean age of the boys and girls was 9.49 ± 5.41 and 10.8 ± 5.64 years, respectively. A CKD-related cause was determined at the age of 4.25 ± 5.11 years on average.

A total of seven boys and seven girls received RRT. The mean age of children receiving RRT was (9.3 ± 5.46). Ten children had a transplanted kidney. The mean age of children at diagnosis of CKD in both these groups was 4.1 years.

CAKUT was the most common cause of CKD (37.9%), but there was no significant difference between the cause of CKD and gender, age, age at diagnosis, and physical development (Table 4).

Analysis of the frequency of clinical signs and association with CKD stage showed that children with CKD stages 4 or 5 (except children after a kidney transplant) had hypertension (87.5%) and anemia (50%) significantly more often and they were prescribed erythropoietin more frequently (37.5%) than those with CKD stages 2 or 3 (Table 5 and Table 6). Growth retardation was more common among children with CKD stages 2–3 and metabolic acidosis was more common among children with CKD stages 4 or 5, but without a significant difference.

Hypertension was diagnosed in 40 (60.6%) children with CKD (transplanted patients included). In children with hypertension, CKD was diagnosed at a younger age than in those without hypertension (median (IQR), 0.9 (0–5.6) vs. 3.9 (0.7–10.3) years, *p* < 0.05) (Figure 1).

Based on the ROC curve analysis (Figure 2), a cut-off value of age when CKD was diagnosed was 2 years. Among children aged >2.0 years, 16 (47.1%) had elevated blood pressure, while among those aged ≤2 years, 24 children had it (75.0%) (*p* = 0.02). Binary logistic regression analysis revealed that children ≤2 years with CKD were at a 3-fold greater risk of having elevated blood pressure (OR = 3.375, 95% CI: 1.186–9.904).

The AUC, sensitivity, and specificity were 67.0%, 60%, and 69.2%, respectively.

## 4. Discussion

### 4.1. Epidemiology of Chronic Renal Failure during 1997–2017

Data on the epidemiology of CKD in the pediatric population are scarce. It is explained by the lack of national registries for this disease across the world. As pointed out by Ahn et al., it is very important as the number of children with CKD is constantly increasing and they develop multiple comorbid conditions such as growth failure, developmental and neurocognitive defects, and impaired cardiovascular health [16]. Moreover, the number of risk factors, such as prematurity or low birth weight, obesity, smoking, hyperuricemia, acute kidney injury, contributing to the development of CKD is increasing [16,17]. The prevalence of pediatric CRF in Lithuania was analyzed for the first time in 1997 [6]. After 10 and 20 years, a significant increase in the prevalence was observed, with the estimated prevalence being as high as 132.1 pmarp in the Lithuanian pediatric population in 2017. Harambat et al. reported that the exact prevalence of CKD stages 2–5 in the EU countries is not known and it depends on the country and ranges from 55–60 to 70–75 pmarp in Spain and Italy, as the disease at its early stages does not show any specific symptoms and is difficult to diagnose [5]. Meanwhile, in the Middle East (Kuwait) and Southeast Asia (Brunei Darussalam) countries, it is considerably greater and accounts for 329 and 736 pmarp, respectively [5,18]. According to the data by the Serbian Pediatric Registry of Chronic Kidney Disease, the prevalence of CKD stages 2–5 is 96.1 pmarp [19]. High prevalence of CKD in the countries of the Middle East and Southeast Asia could be explained by a high number of hereditary diseases in the same family. These data are in agreement with the Lithuanian data, and a greater prevalence of this disease than in other European Union countries could be explained by the declining child population due to increased emigration from post-communist countries and better diagnostics in the earlier stages.

Scarcity of epidemiological data on pediatric CKD as compared to adult CKD could be explained by the lack of a uniform CKD diagnostic and classification system. Only the Kidney Disease: Improving Global Outcomes (KDIGO) guidelines issued in 2002 and updated in 2012 presented the definition and classification of CKD and CRF [4]. Therefore, while comparing the data of our patients during the periods investigated, the issue of CKD classification has arisen. However, our study showed that children with CKD stages 2–4 accounted for the largest patients’ group and this is in line with the data by other authors reporting that the prevalence of CKD stages 2–4 was even 2.4 times greater than that of CKD stage 5 [19]. This is a good indicator of early recognition and better diagnostics of the disease.

Despite these limitations, the incidence of pediatric CKD in Europe is reported to be around 11–12 pmarp for stages 3–5, while the prevalence is ~55–60 pmarp [20]. The prevalence of ESRD during the periods investigated was increasing, and in 2017, it exceeded the EU average reported by the ESPN/ERA-EDTA registry in 2011 by as much as two times and was similar to that of Iceland, Malta, and the United Kingdom [12]. A considerable prevalence of ESRD in Lithuania could be partly explained by timely diagnosis of end-stage CKD and an increasing number of specialized pediatric nephrologists and better care so that in 2017, based on the ESPN data, it was 66.0 per million child population in Lithuania [21].

In many developed countries, the leading cause of CKD is CAKUT [2,5]. Our findings are in agreement with the data of the United States, Italian, Belgian, and Serbian CKD registries, where CAKUT accounted for 48%, 58%, 59%, and 58% of all CKD causes, respectively [5,19]. According to the above registries, cystic kidney diseases were a less common cause of CKD than hereditary nephropathy with the corresponding prevalence rates of 5% vs. 10% in the United States and 9% vs. 19% in Belgium. Our data confirmed this observation: in Lithuania during the 20-year period, the prevalence of cystic kidney diseases also decreased as they are better differentiated from the cysts occurring due to the profound loss of renal tissue in end-stage renal disease. We speculate that higher ranges of CACUT and other hereditary nephropathies worldwide can be associated with increased renal development impairment in embryogenesis.

Evaluation of the age of children with CKD revealed that the mean age was 10 years at all timepoints with the predominance of the male gender (the male-to-female ratio ranged from 1:1.05 to 1:1.32) and this is consistent with the literature data [5,19]. It might be caused by the fact that the main cause of CKD—CAKUT—was more frequently identified among boys.

### 4.2. Epidemiology of CKD in 2017

CKD is a constantly progressing disease leading to ESRD when a child needs immediate RRT or a kidney transplant. Analysis of children with CKD in Lithuania on 1 January 2017 showed that children with CKD stages 2–4 made up the largest percentage, being as high as 78.8% of all CKD patients. The mean age of children at the identification of CKD cause was 4.2 years. Moreover, children receiving RRT were younger than those not receiving RRT. This shows, as confirmed by our data as well, that congenital and hereditary diseases of the kidney are the most common causes of CKD, and many efforts of a multidisciplinary team will be needed in order to identify and treat all complications of CKD early and to promote growth and development, cognitive functions, and integration of these children into the peers’ community.

The frequency of clinical CKD signs depends on the disease stage. Atkinson et al. reported that 73%, 87%, and >93% of children with CKD stages 3, 4, and 5, respectively, were at risk of anemia, showing that the risk of anemia increased with advancing CKD stage [22]. Furthermore, >20% and >40% of children with CKD stages 4 and 5, respectively, treated with erythropoiesis-stimulating agents (ESA) had persistently low hemoglobin levels. Our study confirmed this and showed that anemia was more prevalent among children with CKD stages 4 or 5, and these children were prescribed ESA more frequently than their counterparts with CKD stages 2 or 3. This is very important for the production of erythrocytes in the bone marrow [8].

In the literature, hypertension is recognized as one of the most difficult-to-control complications of CKD [23]. Based on the data of the North American (USA and Canada) Chronic Kidney Disease in Children (CKiD) registry, hypertension was documented in 54% of the children with CKD and as many as 36% of the children who were prescribed anti-hypertensive therapy still demonstrated increased blood pressure [23]. The results of our study confirmed this: hypertension was diagnosed in 60% of children and was significantly more common among those who had CKD stages 4 or 5. Furthermore, we showed that hypertension occurred more frequently in children with CKD when the disease was diagnosed at a younger age. Difficult-to-control hypertension is one of the risk factors for the progression of CKD resulting in cardiovascular lesions and is one of the most common causes of death among children with CKD [23,24].

However, children’s growth and development of cognitive functions are mostly affected [9,25]. Growth retardation has been directly linked to metabolic acidosis and secondary hyperparathyroidism. Based on the data of the CKiD registry, children with short stature before transplantation had a 40% shorter transplant survival time after kidney transplantation [23]. In Lithuania, 16.7% of children with CKD had short stature, with a greater percentage of children being at CKD stages 2 or 3. This could be explained by good medical care of these children—acidosis correction and treatment of secondary hyperparathyroidism and treatment with growth hormones.

Previous studies showed that the manifestation of clinical CKD signs depends not only on the stage of CKD, but also on its cause. Children with hereditary and congenital diseases of the kidney more often develop growth retardation and renal osteodystrophy than children with glomerulonephritis, hypertension, and obesity [2,23]. CAKUT and hereditary nephropathies were the leading causes of CKD in our pediatric cohort, but differences in clinical data between them and other causes of CKD were not found. This could be explained by the small number of patients in our study.

## 5. Conclusions

The number of children with CKD remains stable through decades in Lithuania; however, the prevalence of CKD has increased due to a decline in the pediatric population. CAKUT remains the main cause of CKD. During the period investigated, the proportion of children with hereditary nephropathy was increasing, while that with cystic kidney disease was decreasing. Among children with CKD stages 4 or 5, there were more children with hypertension and anemia. Children ≤2 years with CKD were at a 3-fold greater risk of having elevated blood pressure.

## Figures and Tables

**Figure 1 medicina-57-00112-f001:**
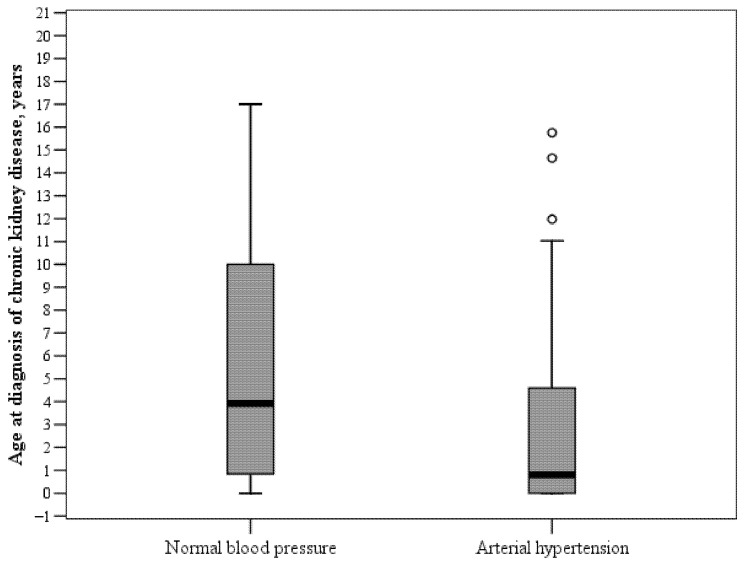
Association between hypertension and age at diagnosis of chronic kidney disease.

**Figure 2 medicina-57-00112-f002:**
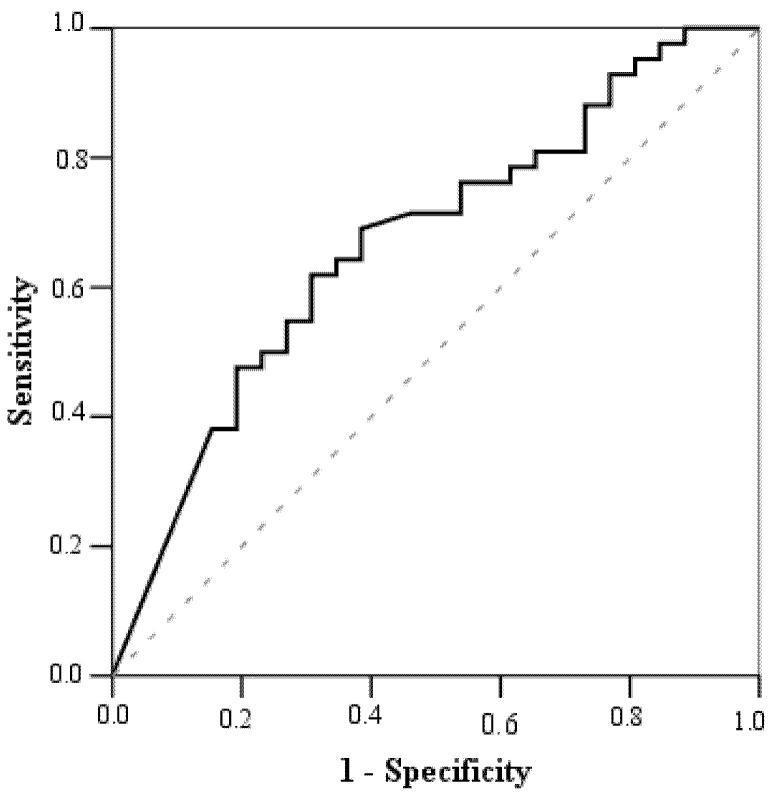
ROC curve to determine a cut-off value of age at diagnosis of CKD to predict hypertension. A cut-off value of age when CKD was diagnosed was 2 years.

**Table 1 medicina-57-00112-t001:** Characteristics of children with CKD by year.

Characteristic	1997 (*n* = 41)	2006 (*n* = 65)	2017 (*n* = 66)	Statistics
Age, mean (SD), years	10.8 (3.5)	10.8 (4.9)	10.3 (5.5)	*F* = 0.75, *df* = 2, *p* = 0.51 *
Gender, *n* (%)				
Girls	20 (48.8)	30 (46.2)	28 (43.1)	*χ*^2^ = 0.342, *df* = 2, *p* = 0.843
Boys	21 (51.2)	35 (53.8)	37 (56.9)	

* ANOVA test; *χ*^2^: chi-squared test.

**Table 2 medicina-57-00112-t002:** Causes of chronic kidney disease by year.

Primary Renal Disease	1997	2006	2017
CAKUT	12 (30.0)	22 (33.8)	25 (37.9)
Cystic kidney disease	13 (32.5) *^,^**	9 (13.8)*	8 (12.1) **
Glomerular diseases	4 (10.0)	11 (16.9)	5 (7.6)
Hereditary nephropathy/metabolic disease	4 (10.0) *^,^**	15 (23.1)*	20 (30.3) **
Thrombotic microangiopathy	3 (7.5)	7 (10.8)	3 (4.5)
Other	4 (10.0)	1 (1.5)	5 (7.6)

Values are numbers (percentage). * *p* < 0.05, comparing 1997 with 2006; ** *p* < 0.05, comparing 1997 with 2017. CAKUT: congenital anomalies of the kidney and urinary tract.

**Table 3 medicina-57-00112-t003:** Distribution of children with chronic kidney disease and children’s characteristics by stage (data for 2017).

CKD Stage	Girls, *n* (%)	Boys, *n* (%)	Age, Median (IQR), Years	Age at Identification of CKD Cause, Median (IQR), Years
2	12 (42.9)	16 (57.1)	11.0 (4.4–13.7) *	2.8 (0–8.7)
3	8 (40.0)	12 (60.0)	8.5 (5.0–15.6) **	0.9 (0–4.3)
4	1 (25.0)	3 (75.0)	3.9 (1.2–11.5) ***	1.5 (0–9.0)
5	7 (50.0)	7 (50.0)	15.2 (11.6–17.3) *^,^**^,^***	4.3 (0.3–8.3)
	*p* = 0.831		*p* = 0.026; *^,^**^,^*** *p* < 0.05	*p* = 0.675

CKD: chronic kidney disease; * *p*: comparison between CKD stage 2 and stage 5; ** *p*: comparison between CKD stage 3 and stage 5; *** *p*: comparison between CKD stage 4 and stage 5.

**Table 4 medicina-57-00112-t004:** Distribution of children with chronic kidney disease and children’s characteristics by the cause of CKD (data for 2017).

Primary Renal Disease (*n*)	Girls, *n* (%)	Boys, *n* (%)	Age, Mean (SD)	Age at Diagnosis, Mean (SD)	Height < 3‰, *n* (%)
CAKUT (25)	11 (44.0)	14 (56.0)	11.3 (5.18)	5.4 (5.66)	3 (12.0)
Cystic kidney disease (8)	2 (25.0)	6 (75.0)	9.0 (5.53)	3.4 (5.94)	2 (28.6)
Glomerular diseases (5)	3 (60.0)	2 (40.0)	13.6 (4.23)	4.4 (3.71)	2 (40.0)
Hereditary nephropathy (20)	10 (50.0)	10 (50.0)	8.6 (5.74)	3.2 (4.46)	4 (20.0)
Thrombotic microangiopathy (3)	1 (33.3)	2 (66.7)	11.7 (6.41)	2.1 (1.49)	0 (0.0)
Other (5)	1 (20.0)	4 (80.0)	10.3 (5.45)	5.4 (6.47)	0 (0.0)

CKD: chronic kidney disease; CAKUT: congenital anomalies of the kidney and urinary tract. *p* > 0.05.

**Table 5 medicina-57-00112-t005:** Clinical symptoms of chronic kidney disease by stage (transplanted patients excluded) (data for 2017).

Clinical Symptoms	CKD Stage, *n* (%)	
	2–3 (*n* = 48)	4–5 (*n* = 8)
Anemia	17 (35.4) *	4 (50.0) *
Hypertension	25 (52.1) **	7 (87.5) **
Acidosis	18 (37.5)	5 (62.5)
Height (< 3‰)	7 (14.9)	1 (12.5)

CKD: chronic kidney disease. *,** *p* < 0.05.

**Table 6 medicina-57-00112-t006:** Treatment of chronic kidney disease and therapy administered by stage (transplanted patients excluded) (data for 2017).

Treatment	CKD Stage, *n* (%)	
	2–3 (*n* = 48)	4–5 (*n* = 8)
Erythropoietin	8 (16.7) *	3 (37.5) *
Sodium hydrocarbonate	18 (37.5) **	6 (75.0) **
Treatment of secondary hyperparathyroidism (alfacalcidol, phosphate binders)	19 (39.6)	6 (75.0)

CKD: chronic kidney disease. *,** *p* < 0.05.

## Data Availability

The data presented in this study are available on request from the corresponding author. The data are not publicly available due to ethical restrictions and data protection policies.

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
