# Peer review of "Epidemiology of Chronic Kidney Disease in Children: A Report from Lithuania"

_medicina, 2021, doi:10.3390/medicina57020112_

Round 1

Reviewer 1 Report

The authors addressed nearly all the raised issues from my first review. 

I have no other objection to the publication of this paper in its current form

Best regards

Author Response

We are very grateful to the reviewer for the comments and good evaluation of our manuscript.

Reviewer 2 Report

Reviewer’s comments and suggestions

The goal of this article was to report the Epidemiology of chronic kidney disease (CKD) in children in Lithuania. The study determined the causes, stage, prevalence, and clinical signs of CKD and demand for renal replacement therapy among Lithuanian children in 2017 and furthermore compared the CKD data with the previous year’s i.e 1997 and 2006. The data of 172 Lithuanian children who had a diagnosis of CKD in 1997 (n = 41); in 2006 (n = 65); and in 2017 (n = 66), were retrospectively analyzed.

The study result reported that prevalence of CKD stages 2–5 was 48.0 pmarp in 1997; 88.7, in 2006; and 132.1, in 2017 (P < 0.01). In 2017, children with CKD stages 4 to 5 (except transplanted children) had hypertension (87.5%) and anemia (50%) (P < 0.01).

Finally, the study concluded that there was no change in the number of children with CKD in Lithuania, and its prevalence increased due to the reduced pediatric population. Children with CKD stages 4 to 5, there were more children having hypertension and anemia. The study was nicely written and understandable, although some minor comments are suggested below that can be incorporated in the revised version.

Decision: Minor revision

  1. Line 19, RRT first time use this abbreviation so it should be in full form.

  1. Pmarp, It is needed to describe the abbreviation and discuss it in the main manuscript.

  1. Line 99, Please mention the numerical values.

  1. What does it indicate, please explain it just for the sake of reader *,**,***P (in table 3)

  1. Figure 1 should be explained in a better way

  1. It is better to explain the ROC curve rather than describing directly to the result. Line 203, Legend should be more explored

  1. Please check the manuscript once again as I find some information required to cite references such as 305-306.

Author Response

We are very grateful to the reviewer for the comments as well as time and efforts to make this study sounder.

Here are our answers to your questions:

  1. We added RRT abbreviation.

  2. Pmarp abbreviation is describes in “Introduction” part of main manuscript, line 7.

  3. Numerical values are indicated in line 106.
  4. We added explanations of P in table 3.

  5. We explained and changed Figure 1.

  6. Explanation after ROC curve was made.

  7.  Thank you for your remark. We checked the manuscript once again.

Reviewer 3 Report

Dear Editor,

Thank you for asking me the review process. The manuscript entitled “Epidemiology of chronic kidney disease in children” is very interesting to publish in Medicina. The author analyzed the prevalence of chronic kidney disease (CKD) in pediatric population. They claimed the prevalence of CKD increased due to reduced pediatric population. They need explain the reasons why the prevalence of KCD increased even if the children are decreased. They also claimed that the main causes of CKD in pediatric population are congenital anomalies of the kidney and urinary tract (CAKUT). I want to ask the author need to discuss the reason of CAKUT in Lithuania. This will improve the quality of the manuscript.

Thank you

Author Response

We are very grateful to the reviewer for the comments as well as time and efforts to make this study sounder.

We made changes in the manuscript according to your comments and suggestions.

This manuscript is a resubmission of an earlier submission. The following is a list of the peer review reports and author responses from that submission.

Round 1

Reviewer 1 Report

A correct and complete article is presented. However, I believe that what it brings should be highlighted. That is, its usefulness and, therefore, its necessity.

Introduction. I think it could be completed. Above all, to clearly state what a work with this objective brings. In other words, it should be more justified. It is not clear what this work contributes, and therefore whether it should be carried out.

Materials and methods. The reasons for the choice of the years selected for the retrospective analysis (1997, 2006 and 2017) could be clarified. On the other hand, we are ending 2020. The procedure used to obtain the data is not clearly described. This point would be better described in a broader and more specific way.

Discussion. Three years have passed since the study was conducted. I wonder what the reasons are for this delay. Could more current data be incorporated? Have protocols been initiated from this study?

Reviewer 2 Report

The authors present epidemiology data of chronic kidney disease in Lithuania. Some issues should be clarified and the text changed :

  1. Was there an overlap in the diseased children between the studied time periods? Or were only new cases included? For example age range in 1997 was 1-16 years, which means that theoretically some of these children have reached year 2006. The same holds through if comparing the age ranges between 2006 and 2017.
  2. Was the Schwarz formula used for calculation of GFR in all studied time periods ?
  3. I suppose that the title of 3.1 refers to 1997 (?). If so table 1 which compares all time periods should be written in the end. Similarly, Figure 1 must also include children of the 1997 period, so that the reader can easily compare the CKD stages.
  4. Table 2 shows data of the 2017 time period but the characteristics of cases are described in paragraph 3.2. and summarized in Table 3. I wonder, why the authors give extended details only for 2017. Where there more data available?
  5. The structure and the order of tables must be changed. For example tables 4-6 and Figure 2 refer to 2017?
  6. Which was proportion of children needing RRT in each time period and which modality was preferred?
  7. Which was the proportion of children transplanted in each time period and what was the outcome ?
  8. Are the any survival data of the children with and without transplantation available?
  9. C-Statistics are used only for 2017. I wonder what is their usefulness and why this was not done for the earlier time periods.